# Multigene typing of *Giardia Duodenalis* isolated from tuberculosis and non-tuberculosis subjects

Hanieh Mohammad Rahimi[1], Ehsan Javanmard[2], Ali Taghipour[3,4], Ali Haghighi[5], Hamed Mirjalali [ID][1] *

1 Foodborne and Waterborne Diseases Research Center, Research Institute for Gastroenterology and Liver Diseases, Shahid Beheshti University of Medical Sciences, Tehran, Iran, 2 Dept. of Medical Parasitology and Mycology, School of Public Health, Tehran University of Medical Sciences, Tehran, Iran, 3 Department of Medical Parasitology and Mycology, School of Medicine, Jahrom University of Medical Sciences, Jahrom, Iran, 4 Zoonoses Research Center, Jahrom University of Medical Sciences, Jahrom, Iran, 5 Department of Medical Parasitology and Mycology, Faculty of Medicine, Shahid Beheshti University of Medical Sciences, Tehran, Iran

* hamedmirjalali@sbmu.ac.ir, hamed_mirjalali@hotmail.com

**Data Availability Statement:** All relevant data are within the paper. Generated sequences were

## Abstract

*Giardia duodenalis* is a cryptic protozoan, which has eight assemblages (A-H). Assemblages A and B are the main genotypes reported from humans with probable anthroponotic and zoonotic transmission. The current study aimed to characterize *G. duodenalis* assemblages in tuberculosis (TB) patients and healthy subjects using multilocus genotyping (MLG). Thirty *Giardia*-positive stool samples, which were obtained from TB patients and healthy subjects were included in the study. After total DNA extraction, three β-giardin *(bg)*, triosephosphate isomerase *(tpi)*, glutamate dehydrogenase *(gdh)* genes were amplified and sequenced. Obtained sequences were compared to the GenBank database to characterize assemblages. Phylogenetic analysis using Maximum Likelihood (ML) and Tamura 3-parameter was performed for each gene. From 30 *Giardia*-positive subjects, 17 (57%) and 13 (43%) were from healthy and TB-infected subjects, respectively. There was no significant co-existence of *Giardia* and tuberculosis (*P*-value = 0.051). In addition, 14 (46.7%) and 16 (53.3%) of *Giardia* isolates were from asymptomatic and symptomatic subjects, respectively. PCR amplification was successful in 25 single samples (83.3%) consisted of 20 for *tpi*, 15 for *bg*, and 13 for *gdh* genes. Accordingly, 13/25 (52%) and 8/25 (32%) belonged to assemblage A and assemblages B, respectively, whereas 4/25 (16%) were either assemblage A or B with different genes at the same time. Significant correlation between assemblages and TB, age, and symptoms was not seen. The phylogenetic analyses represented no separation based on TB and gastrointestinal symptoms. Assemblage A was the predominant genotype in samples. The high frequency of assemblage AII indicated importance of anthroponotic transmission of *Giardia* in both healthy and TB patients. In addition, considering the exclusive reports of sub-assemblage AIII in wild ruminants, the presence of AIII in the current study have to be carefully interpreted. The inconsistency between the assemblage results of either *bg* or *gdh* loci with *tpi* gene

submitted to the GenBank database under accession numbers OM115964 to OM115983, OM115984 to OM115998, and OM115999 to OM116011, for the tpi, bg, and gdh loci, respectively.

**Funding:** This study was financially supported by the Research Institute for Gastroenterology and Liver Diseases, Shahid Beheshti University of Medical Sciences with grant number: RIGLD-1143. The funders had no role in study design, data collection and analysis, decision to publish, or preparation of the manuscript. Hamed Mirjalali, Ali Haghighi, and Hanieh Mohammad Rahimi receive salary from Shahid Beheshti University of Medical Sciences.

**Competing interests:** The authors have declared that no competing interests exist.

signifies the insufficiency of single gene analysis and the necessity for MLG in molecular epidemiology of *G. duodenalis*.

## Introduction

*Giardia duodenalis* (syn. *Giardia intestinalis*, *Giardia lamblia*) is a cosmopolitan flagellated protozoan, which is reported from not only humans, but also a broad spectrum of animals. In addition to *G. duodenalis*, there are at least seven other species including *G. agilis*, *G. ardeae*, *G. psittaci*, *G. muris*, *G. microti*, *G. peramelis*, and *G. cricetidarum*, which colonize the intestine of a broad variety of animals from amphibians to birds [1]. Unlike other species, *G. duodenalis* is not limited to a specific host, and is reported from humans and domesticated/wild animals, which increases the probability of zoonotic transmission [2]. The main transmission rout of *Giardia* spp., in humans is fecal-oral via ingestion of infective cysts defecated by either humans or animals, as well as contaminated food and water [3–5].

Although infection by *G. duodenalis* in humans is mostly asymptomatic, variety of gastrointestinal manifestations such as bloating, nausea, flatulence, fatigue, weight loss, diarrhea, and steatorrhea might be reported [6–8]. The correlation between diarrhea, as the main symptom of giardiasis, and colonization of *Giardia* has been controversial [9, 10]. However, it is suggested that the presence of *G. duodenalis* may protect infected subjects from diarrhea due to other intestinal pathogens [6].

Despite the similar morphology, eight assemblages (A-H) has been characterized based on the molecular diversity within *G. duodenalis* genomes. Accordingly, assemblages A and B are the major reported genetic lineages from humans and animals, while other six assemblages are exclusively reported from animals [11, 12]. Nevertheless, the majority of assemblages A and B in humans and high occurrence of host-adapted assemblages in companion animals suggest that the zoonotic transmission of *G. duodenalis* is less common than expected before [11].

To study the molecular characterizations and phylogenetic relationship of genetic lineages of *G. duodenalis*, several genetic markers have been investigated, so far. Accordingly, β-giardin (*bg*), triosephosphate isomerase (*tpi*), glutamate dehydrogenase (*gdh*), internal transcribed spacer (*its*), 18S ribosomal RNA (*ssu rRNA)*, and elongation factor (*ef*) are candidates, in which the first three genetic markers are commonly employed for assemblage characterization of *G. duodenalis* [13–15].

The presence and assemblage distribution of *G. duodenalis* have been evaluated among different groups of healthy subjects and persons with background diseases. However, there are limited data about the assemblages of *G. duodenalis* in patients who suffer from tuberculosis (TB) [16–19]. Indeed, there is no data about distribution of assemblages in TB patients. Therefore, the current study aimed to characterize assemblages of *G. duodenalis* isolated from human subjects with and without TB.

## Materials and methods

### Ethics approval and consent to participate

All procedures in this study were according to the received approval from the Ethics Committee of the Shahid Beheshti University of Medical Science (SBMU), Tehran, Iran (IR.SBMU. MSP.REC.1395.323) and the Ethical Review Committee of the Research Institute for Gastroenterology and Liver Diseases, Shahid Beheshti University of Medical Sciences, Tehran, Iran (IR. SBMU.RIGLD.REC.1399.056).

Consent was informed to participates and verbal consent was obtained from all subjects and/or their legal guardian(s). For those patients with age≤16, informed consent was obtained from their respective parent(s)/guardian(s) as well.

## Stool sample collection and DNA extraction

This study was conducted on 30 *Giardia*-positive samples, which had been obtained from 427 stool samples collected from our previous studies (261 TB patients) [19, 20], as well as samples, which were referred to the Foodborne and Waterborne Diseases Research Center, Research Institute for Gastroenterology and Liver Diseases, Shahid Beheshti University of Medical Sciences. All samples were either positive or suspected for *G. duodenalis* using lugol's iodine staining and microscopically examination. Total DNA was extracted from stool samples using DNA stool extraction mini kit (Yekta Tajhiz, Tehran, Iran) with some modifications [21]. Purified DNA was stored at -20˚C.

## Multilocus genotyping

Nested PCR was employed to amplify the *bg*, *gdh*, and *tpi* genes among *G. duodenalis* isolates using primers and PCR conditions, which were mentioned elsewhere [22] (Table 1). To characterize assemblages of each isolate, PCR products were sequenced. Raw sequence data in forward direction was viewed using the Chromas Lite version 2.6 sequence analysis program (https://chromas.software.informer.com/2.6/). The nucleotides were checked and manually edited, where required. The BLAST tool (http://blast.ncbi.nlm.nih.gov/Blast.cgi) was used to compare nucleotide sequences with sequences previously submitted to the GenBank database.

## Phylogenetic analyses

Generated sequences were trimmed and aligned based on appropriate reference sequences using BioEditv.7.2.6 software. Phylogenetic trees were constructed for targeted fragments of each *tpi*, *bg*, and *gdh* gene of *G. duodenalis* using the Maximum-likelihood algorithm and Tamura 3-parameter model in MEGAX software, together with a number of sequences, which were retrieved from the GenBank database to evaluate the molecular distance and phylogenetic

**Table 1. Targets and primers used for PCR amplification of *gdh*, *tpi* and *bg* genes of *G. duodenalis*.**

| Genetic loci | Primer name | Primers sequence (5′ - 3′) | Size (bp) | Annealing temperature | Ref |
|---|---|---|---|---|---|
| *tpi* | AL3543* | AAATIATGCCTGCTCGTCG | 605 | 51 | [22] |
| | AL3546* | CAAACCTTITCCGCAAACC | | | |
| | AL3544 | CCCTTCATCGGIGGTAACTT | 532 | 55 | |
| | AL3545 | GTGGCCACCACICCCGTGCC | | | |
| *gdh* | Ghd1* | TTCCGTRTYCAGTACAACTC | 754 | 55 | |
| | Ghd2* | ACCTCGTTCTGRGTGGCGCA | | | |
| | Ghd3 | ATGACYGAGCTYCAGAGGCACGT | 530 | 58 | |
| | Ghd4 | GTGGCGCARGGCATGATGCA | | | |
| *bg* | G7* | AAGCCCGACGACCTCACCCGCAGTGC | 753 | 51 | |
| | G759* | GAGGCCGCCCTGGATCTTCGAGACGAC | | | |
| | GiarF | GAACGAACGAGATCGAGGTCCG | 511 | 55 | |
| | GiarR | CTCGACGAGCTTCGTGTT | | | |

* Primers used in the first PCR step.Edited sequences, and their resulting amino acid sequences, were submitted to the GenBank database under accession numbers: OM115964 to OM115983, OM115984 to OM115998, and OM115999 to OM116011, for the *tpi*, *bg*, and *gdh* loci, respectively.

relationships among isolates [23]. The reasons for choosing Tamura 3-parameter are the analyzing both transitional and transversional rates, G+C content bias, and correcting multiple hits (http://www.megasoftware.net/) [23]. The reliabilities of the trees were assessed using the bootstrap analysis with 1000 replications.

### Statistical analysis

Statistical analyses, Pearson's Chi-square ($\chi2$) for independence and Fisher's exact tests incorporated in SPSS version 23 software (SPSS Inc. Chicago, IL, USA) were employed to compare the frequency of *G. duodenalis* assemblages in symptomatic and asymptomatic subjects. Statistical significance was set as a *P*-value < 0.05.

## Results

### Prevalence and clinical data

From 427 tested stool samples, 227 (53.16%) and 200 (46.84%) were male and females, respectively. Indeed, 261 (61.12%) patients were TB positive and 166 (38.88%) were healthy people (non-TB group). From these samples, 30 (7.02%) were positive or suspected for *Giardia* using microscopy analysis including 17 (57%) and 13 (43%) for healthy and TB-infected subjects, respectively. A significant co-existence of *Giardia* and tuberculosis was not seen (*P*-value = 0.051). From *Giardia*-positive samples, 17 (57%) were males and 13 (43%) were females. There was no statistically significant correlation between the presence of *Giardia* and gender (*P*-value = 0.709). The mean age ± standard deviation (SD) and the median age of *Giardia*-positive subjects were 28.27±18.13 and 29, respectively. The age range of positive samples were from < 7 to 65 years. The highest frequency of *Giardia* infection was observed in the age group of 8–20 (26.6%; 8/30), while the age groups of <7 and 51–65 showed the lowest frequency (13.3%; 4/30). In addition, 14 (46.7%) of infected subjects were asymptomatic and 16 (53.3%) showed clinical symptoms including abdominal pain with diarrhea 12 (75%), diarrhea 3 (18.75%), and nausea and vomiting 1 (6.25%) (Tables 2 and 3).

### Molecular detection and genotyping

From 30 microscopically *Giardia*-positive samples, PCR products of all three genes was successfully amplified in 25 single samples (83.3%) consisted of 20, 15, and 13 for *tpi*, *bg*, and *gdh* genes, respectively. The sequence results of three *bg* and seven *gdh* genes showed high

**Table 2. The correlation between demographic data and assemblages.**

| Demographic data | | A | B | NA | Total | *P-value* |
|---|---|---|---|---|---|---|
| **Gender** | Male | 9 | 5 | 3 | 17 | 0.887 |
| | Female | 6 | 5 | 2 | 13 | |
| **Tuberculosis** | No | 8 | 7 | 2 | 17 | 0.626 |
| | Yes | 7 | 3 | 3 | 13 | |
| **Presence of Symptom** | Yes | 9 | 6 | 1 | 16 | 0.304 |
| | No | 6 | 4 | 4 | 14 | |
| **Clinical manifestations** | Diarrhea | 3 | 0 | 0 | 3 | 0.398 |
| | Diarrhea/Abdominal pain | 5 | 6 | 1 | 12 | |
| | Nausea & Vomiting | 1 | 0 | 0 | 1 | |
| | No symptoms | 6 | 4 | 4 | 14 | |

**Note:** A: assemblage A; B: assemblage B; NA: not assigned (those samples which were amplified by none of three genes).

**Table 3. The correlation between assemblages and age groups.**

| Assemblages | Age groups | | | | | Total | *P-value* |
|---|---|---|---|---|---|---|---|
| | <7 | 8–20 | 21–35 | 36–50 | 51–65 | | |
| A | 1 | 4 | 3 | 6 | 1 | 15 | 0.204 |
| B | 3 | 3 | 3 | 0 | 1 | 10 | |
| NA | 0 | 1 | 1 | 1 | 2 | 5 | |
| Total | 4 | 8 | 7 | 7 | 4 | 30 | |

**Note:** NA: not assigned (those samples which were amplified by none of three genes).

similarity to either bacteria or viruses. Consensus assemblage analysis showed that 13/25 (52%) and 8/25 (32%) were identified as assemblage A and assemblages B, respectively, whereas 4/25 (16%) were either assemblage A or B with different genes.

The *bg* gene analysis showed that 8/15 (53%) of isolates belonged to assemblage A with sub-assemblages AII and AIII, and 7/15 (47%) isolates belonged to assemblage B with sub-assemblage BIII. Result of the *gdh* gene showed that 7/13 (54%) were identified as assemblage A with sub-assemblages AII and AIII, and 6/13 (46%) were identified as assemblage B with sub-assemblages BIII and BIV. From 20 successful sequences for *tpi* gene, 13 (65%) belonged to assemblage A with sub-assemblage AII, and 7 (35%) were assemblage B with sub-assemblages BIII. Overall, AII was the most prevalent sub-assemblage detected in 8/25 (32%), followed by BIII in 5/25 (20%), AIII in 1/25 (4%), and BIV in 1/25 (4%). Non-consensus sub-assemblages were seen in five samples including AII/AIII in 4/25 (16%) and BIII/BIV in 1/25 (4%). In addition, non-consensus assemblages/sub-assemblages AII/BIII and BIII/BIV/AII were characterized in 3/25 (12%) and 1/25 (4%), respectively (Table 4).

Among TB patients, from 13 *Giardia*-positive samples, assemblages A and B were characterized among seven (53.84%) and three (23.07%), respectively, while three (23.07%) of remained samples did not amplified or were failed in sequencing. From 17 *Giardia*-positive samples, Assemblages A and B were characterized among six (35.29%) and five (29.41%) of non-TB subjects, respectively, while four (23.53%) samples were assemblage A or B with different genes and two (23.07%) did not amplified or were failed in sequencing.

## Phylogenetic analyses

Phylogenetic analysis of *bg*, *tpi*, and *gdh* genes revealed that assemblages A and B were clearly separated and grouped with reference assemblages retrieved from the GenBank database for each gene (Fig 1). Moreover, the phylogenetic analysis of sequences of all three genes represented that there was no separation based on the presence of TB, gastrointestinal symptoms, sources, and geographical areas.

## Discussion

*Giardia duodenalis* is a prevalent protozoan, particularly in developing region, which infects humans and a board range of animals [13]. The prevalence of *G. duodenalis* in Iran has been even reported more than 30%; however, the prevalence rate may vary regarding the studied population and employed diagnostic methods [24–28]. Although among epidemiological surveys, *G. duodenalis* is one of the most frequently reported protozoan, our knowledge is still insufficient about the molecular epidemiology and circulating assemblages of the parasite in Iran. The current study is one of the rare research mining the molecular characterization and assemblages of *G. duodenalis* in TB patients using multilocus genotyping (MLG) in the world.

**Table 4. Demographic characteristics and clinical symptoms of TB patients and non-TB group, *G. duodenalis*-positive isolates, and the relevant genotypes.**

| No. | Gender | Age | Job | Symptoms | Background diseases | Assemblage/Acc No. | | | Assigned genotype |
|-----|--------|-----|-----|----------|---------------------|-------|-------|-------|-------------------|
| | | | | | | *gdh* | *bg* | *tpi* | |
| GH1 | Female | 11 | Farmer | Diarrhea/ Abdominal Pain | No Disease | NA | AIII (OM115984) | AII (OM115964) | AIII/AII |
| GH2 | Male | 5 | Ranchman | Diarrhea/ Abdominal Pain | No Disease | AII (OM115999) | AII (OM115985) | BIII (OM115965) | AII/BIII |
| GH3 | Female | 7 | Farmer | Diarrhea/ Abdominal Pain | No Disease | B (OM116000) | BIII (OM115986) | BIII (OM115966) | BIII |
| GH4 | Female | 8 | Farmer | Diarrhea/ Abdominal Pain | No Disease | BIV (OM116001) | BIII (OM115987) | AII (OM115981) | BIII/BIV/AII |
| GH5 | Male | 7 | Ranchman | Diarrhea/ Abdominal Pain | No Disease | NA | NA | BIII (OM115967) | BIII |
| GH6 | Male | 10 | Ranchman | Diarrhea/ Abdominal Pain | No Disease | A (OM116002) | AIII (OM115988) | AII (OM115968) | AII/AIII |
| GH7 | Female | 10 | Farmer | Diarrhea/ Abdominal Pain | No Disease | BIV (OM116003) | BIII (OM115989) | BIII (OM115969) | BIII/BIV |
| GH8 | Male | 14 | Farmer | Diarrhea/ Abdominal Pain | No Disease | NA | NA | NA | NA |
| GH9 | Female | 14 | Farmer | Diarrhea/ Abdominal Pain | No Disease | AII (OM116004) | AIII (OM115990) | AII (OM115970) | AII/AIII |
| GH10 | Male | 7 | Farmer | Diarrhea/ Abdominal Pain | No Disease | B (OM116005) | BIII (OM115991) | NA | BIII |
| GH11 | Male | 9 | Farmer | Diarrhea/ Abdominal Pain | No Disease | A (OM116006) | AII (OM115992) | BIII (OM115971) | AII/BIII |
| GH12 | Male | 11 | Farmer | Diarrhea/ Abdominal Pain | No Disease | B (OM116007) | BIII (OM115993) | AII (OM115972) | AII/BIII |
| GH13 | Female | 29 | Farmer | No Symptoms | No Disease | NA | NA | BIII (OM115973) | BIII |
| GH14 | Female | 28 | Farmer | No Symptoms | No Disease | AII (OM116008) | AIII (OM115994) | AII (OM115974) | AII/AIII |
| GH15 | Male | 29 | Ranchman | No Symptoms | No Disease | AII (OM116009) | AII (OM115995) | NA | AII |
| GH16 | Female | 31 | Farmer | No Symptoms | No Disease | NA | NA | AII (OM115975) | AII |
| GH17 | Male | 51 | Farmer | No Symptoms | No Disease | NA | NA | NA | NA |
| GH18 | Male | 65 | Farmer | No Symptoms | TB | BIV (OM116010) | B (OM115996) | NA | BIV |
| GH19 | Male | 44 | Ranchman | No Symptoms | TB | AII (OM116011) | NA | AII (OM115982) | AII |
| GH20 | Male | 43 | Self-employed | Diarrhea | TB | NA | NA | AII (OM115983) | AII |
| GH21 | Female | 58 | Housewife | No Symptoms | TB | NA | NA | NA | NA |
| GH22 | Female | 32 | Housewife | No Symptoms | TB | NA | NA | NA | NA |
| GH23 | Male | 50 | Shepherd | Nausea and vomiting | TB | NA | NA | AII (OM115976) | AII |
| GH24 | Male | 30 | Self-employed | No Symptoms | TB | NA | B (OM115997) | NA | B |
| GH25 | Male | 44 | Farmer | Diarrhea | TB | NA | NA | AII (OM115977) | AII |
| GH26 | Male | 36 | Self-employed | No Symptoms | TB | NA | NA | NA | NA |
| GH27 | Female | 40 | Self-employed | No Symptoms | TB | NA | NA | AII (OM115978) | AII |
| GH28 | Male | 56 | Self-employed | Diarrhea | TB | NA | NA | AII (OM115979) | AII |
| GH29 | Female | 25 | Self-employed | No Symptoms | TB | NA | NA | BIII (OM115980) | BIII |

(*Continued*)

**Table 4.** (Continued)

| No. | Gender | Age | Job | Symptoms | Background diseases | Assemblage/Acc No. | | | Assigned genotype |
|---|---|---|---|---|---|---|---|---|---|
| | | | | | | *gdh* | *bg* | *tpi* | |
| **GH30** | Female | 44 | Housewife | No Symptoms | TB | NA | AIII (OM115998) | NA | AIII |

**Note:** NA: not amplified (in addition, either failed or similar to bacteria and viruses in sequencing); TB: tuberculosis; *gdh*: glutamate dehydrogenase; *bg*: beta giardin; *tpi*: triosephosphate isomerase.

Assemblage characterization of *G. duodenalis* is a challenge, since multi-copy genes, like *ssu rRNA* gene, are not discriminative enough to identify assemblages and sub-assemblages [29, 30]. Besides, success rate for amplification of single copy genes, which are discriminative, varies from 11 to 90% [31]. Therefore, multigene typing is a valid model for molecular characterization of *G. duodenalis* at assemblage and sub-assemblage level [13]. Among reliable genes for assemblage characterization of *G. duodenalis*, three loci *tpi*, *gdh*, and *bg* are well-known genetic targets [13]. In the current study, from 30 microscopically positive samples, 25 samples were amplified by each/two/all genes including 20, 15, and 13 samples for *tpi*, *bg*, and *gdh*, respectively. This observation is in accordance to previously published papers indicating inconsistency between the prevalence of *G. duodenalis*-positive samples in microscopy and molecular amplification [32–34]. This discrepancy could be related to the quality of extracted DNA, the copy number of targeted genes, and the presence of PCR inhibitors [31, 34].

In the current study, three and seven sequences of *bg* and *gdh* genes were highly similar to bacteria (*Bifidobacterium*, *Feacalibacterium*, *Pseudomonas*, *Kinneretia* and *Escherichia coli*) and virus (Siphoviridae), while none of *tpi* locus sequences were identical to non-*Giardia* sequences. Such results highlight the concern for false positive results of non-specific target gene amplification by PCR without sequencing [31].

As a result, assemblage A was the most prevalent genotypes in the current study. Assemblage A is the predominant genetic lineage of *G. duodenalis* in humans in most of molecular

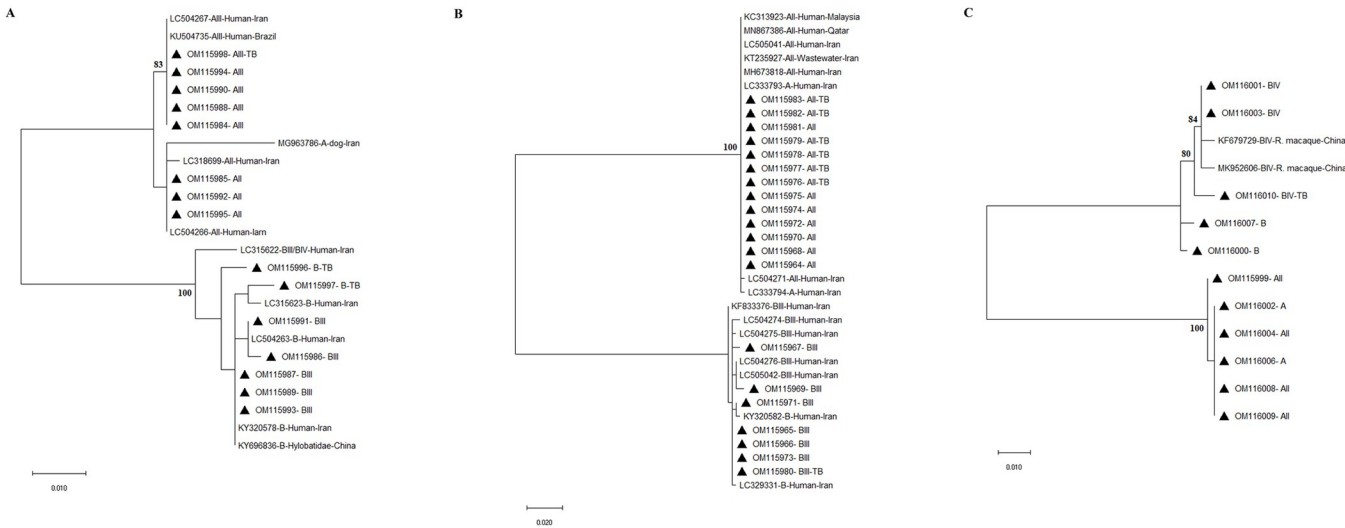

**Fig 1.** Phylogenetic trees of **A)** *bg*, **B)** *tpi*, and **C)** *gdh* genes of *G. duodenalis* obtained from this study together with reference sequences retrieved from GenBank. The trees were constructed based on the Maximum likelihood (ML) method and the Tamura 3-parameter model in MEGAX. Bootstrap values lower than 75 were omitted. Our sequences were indicated with black-filled triangles (▲).

studies in Iran [28, 34, 35]. In a study conducted by Sarkari et al. [36], assemblage A was identified among 74.4% of *Giardia*-positive samples based on the amplification followed by restriction fragment length polymorphism (RFLP) of partial fragment of *gdh* gene. Mahmoudi et al. [28] reported assemblage A as the major genotype based on the partial amplification and sequencing of *gdh* gene. In addition, Bahramdoust et al., [35] designed and evaluated a real-time PCR coupled with high resolution melting curve analysis (HRM) to detect and characterize *G. duodenalis* in humans and dogs, and reported assemblage A as the major genotype in humans. Although assemblage A seems to be the predominant genotype in humans in Iran [37–39], there are controversial results, as well. For example, in a MLG study conducted in Iran, the presence of assemblages A and B were similar [34]. Most of assemblage A and all of assemblage B were AII and BIII/BIV, respectively, which are supposed to be responsible for anthroponotic transmission [2, 34]. Interestingly, an inconsistency between AII and AIII was observed in sub-assemblage analysis by different genes. In addition, all sub-assemblage AIII were identified based on *bg* gene sequencing. However, AIII was supposed to be exclusively reported from wild ruminants [40, 41], and the presence of this sub-assemblage in this study may be attributed to the misassigned sequences, which have been submitting to the GenBank database, and have to be carefully interpreted. Furthermore, it is suggested to employ either/both *tpi* or/and *gdh* genes alongside with *bg* locus for sub-assemblage analysis.

Regarding our results, significant correlation was not seen between clinical symptoms and certain assemblage. Furthermore, there was no correlation between certain assemblage and tuberculosis. Actually, a little is known about the correlation between genetic variability of *Giardia* and presentation of clinical symptoms. Although there are reports demonstrating an association between assemblages and type of symptoms [42, 43], most of studies failed to link genetic variability of *Giardia* with symptoms [34, 44, 45].

Phylogenetic analysis showed that all sequences were clearly separated based on the assigned assemblages and grouped with their reference sequences for each gene. However, an inconstancy was appeared between the assigned assemblages by either *bg* or *gdh* genes and *tpi*. In another word, in four *G. duodenalis* sequences, which were characterized as assemblages either A or B with two *bg* and *gdh* genes, the results of *tpi* gene was conflicting. This issue could be related to the segregation sites, the number of mutation, single nucleotide polymorphisms (SNPs), and discriminatory power of each target gene [13]. This finding highlights the insufficiency of a single target gene screening and the need for MLG investigation for molecular epidemiology studies of *Giardia*.

## Conclusion

This study is the first analyzing *G. duodenalis* assemblages in TB patients using MLG approach. The assemblage A was the predominant genotype in our isolates, but a significant correlation between certain assemblage with symptoms and TB was not observed. The high prevalence of assemblage AII indicated the importance of anthroponotic transmission of *Giardia* in both healthy and TB-infected subjects. In addition, considering the exclusive reports of sub-assemblage AIII in wild ruminants, the presence of AIII in the current study have to be carefully interpreted. The inconsistency between the assemblage results of either *bg* or *gdh* genes and *tpi* gene signify the insufficiency of single gene analysis and the necessity of MLG in molecular epidemiology of *G. duodenalis*.

## Acknowledgments

The authors thank all members of the Foodborne and Waterborne Diseases Research Center for their collaborations.

## Author Contributions

**Conceptualization:** Ali Haghighi, Hamed Mirjalali.

**Investigation:** Hanieh Mohammad Rahimi, Ehsan Javanmard, Ali Taghipour, Hamed Mirjalali.

**Methodology:** Ehsan Javanmard.

**Resources:** Ehsan Javanmard, Ali Taghipour, Ali Haghighi.

**Supervision:** Hamed Mirjalali.

**Validation:** Hamed Mirjalali.

**Visualization:** Hanieh Mohammad Rahimi.

**Writing – original draft:** Ehsan Javanmard, Ali Taghipour, Hamed Mirjalali.

**Writing – review & editing:** Hamed Mirjalali.

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
