## [Decision Letter · Decision Letter 0]

3 Jan 2023

PONE-D-22-33229Multigene Typing of Giardia Duodenalis Isolated from Tuberculosis and Non-Tuberculosis SubjectsPLOS ONE

Dear Dr. Mirjalali,

Thank you for submitting your manuscript to PLOS ONE. After careful consideration, we feel that it has merit but does not fully meet PLOS ONE’s publication criteria as it currently stands. Therefore, we invite you to submit a revised version of the manuscript that addresses the points raised during the review process.

We look forward to receiving your revised manuscript.

Kind regards,

Bibi Razieh Hosseini Farash

Academic Editor

PLOS ONE

“No. The funders had no role in study design, data collection and analysis, decision to publish, or preparation of the manuscript.”

Reviewers' comments:

Reviewer's Responses to Questions

**Comments to the Author**

1. Is the manuscript technically sound, and do the data support the conclusions?

Reviewer #1: Yes

Reviewer #2: Yes

2. Has the statistical analysis been performed appropriately and rigorously? 

Reviewer #1: No

Reviewer #2: Yes

3. Have the authors made all data underlying the findings in their manuscript fully available?

Reviewer #1: Yes

Reviewer #2: Yes

4. Is the manuscript presented in an intelligible fashion and written in standard English?

Reviewer #1: Yes

Reviewer #2: Yes

5. Review Comments to the Author

Reviewer #1: 1. Sample size is not sufficient for statistical analysis. increase the sample size of your study.

2. Results reported in this study is different from authors' previous study (Ref. 19) including sample size and frequency of Giardia lamblia. The number of microscopically positive-Giardia samples in this study was reported 30, while in the previous study 8 samples were reported. Also the sample size of non-TB group is not the same as the previous study.

Reviewer #2: The paper is of interest in the field as scare data are available targeting Giardia coinfection with TB

The work was performed following the standard’s molecular approaches

Below few comments to improve the MS quality:

-English should be revised to avoid typo errors: e.g., L32, L41, L50, L57, L58, L61, L72, L131, L151, L186, L197, L209, L223, L231, L242...

-In methodology part, authors should mention the multiple alignment software used in their analysis

-In methodology part, authors should explain why they chose the "Tamura 3-parameter model" in their phylogenetic analysis

-Legends of Tables 2, 3 and 4 are misplaced

-A footnote for Table 2 is required to understand the reported data

-Higher resolution of the figure needs to be provided

Kind regards

6. PLOS authors have the option to publish the peer review history of their article (what does this mean?). If published, this will include your full peer review and any attached files.

Reviewer #1: No

Reviewer #2: No

---

## [Author Response · Author response to Decision Letter 0]

24 Feb 2023

Dear Dr. Hosseini Farash

Academic Editor

PLOS ONE

Manuscript ID: PONE-D-22-33229

Thank you for allowing me to submit a revised draft of our manuscript titled: “Multigene Typing of Giardia Duodenalis Isolated from Tuberculosis and Non-Tuberculosis Subjects”. We appreciate the time and effort you have dedicated to providing your valuable feedback on our manuscript. We are grateful to the reviewers for their insightful comments on our paper. We have been able to incorporate changes to reflect most of the suggestions provided by the reviewers. We have highlighted in yellow the changes within the manuscript. 

Review Comments to the Author

Reviewer #1: 

Response: We would like to thank reviewer for deep review and valuable comments and revisions. We hope this version could be suitable for publication.

1. Sample size is not sufficient for statistical analysis. increase the sample size of your study.

Response: With respect to reviewer, for this study we analyzed 30 Giardia-positive samples which were isolated from 427 samples during 2 years. We understand your concern, but there is no possibility to extend the number of samples on TB patients at this time.

2. Results reported in this study is different from authors' previous study (Ref. 19) including sample size and frequency of Giardia lamblia. The number of microscopically positive-Giardia samples in this study was reported 30, while in the previous study 8 samples were reported. Also the sample size of non-TB group is not the same as the previous study.

Response: Thank you so much for your valuable note. We sorry for this mistake. The number of samples was 427 not 327. Actually, sampling from TB patients was continued after our previous study (ref.19) to 261 TB samples (at the time of our previous study the number of samples was 161). In addition, in our previous study we just reported “Giardia-positive” for those samples which were definitively microscopically identical, but not for those that were suspected or have changes in their shapes (in this study we extracted DNA from both definitive positive and suspected samples). Therefore, the number of positive samples with molecular methods in 261 TB patients (considering all definitive and suspected Giardia positive by microscopy) was 13.

For healthy subjects, we have not employed positive samples from those healthy subjects which were included in our previous study. Giardia positive samples from non-TB patients was isolated from those apparently healthy subjects who referred to the Foodborne and waterborne Diseases Research Center at the same time. 

Reviewer #2: 

The paper is of interest in the field as scare data are available targeting Giardia coinfection with TB

The work was performed following the standard’s molecular approaches

Response: We would like to thank reviewer for deep review, valuable comments and revisions, and positive feedback. We hope this version could be suitable for publication.

Below few comments to improve the MS quality:

-English should be revised to avoid typo errors: e.g., L32, L41, L50, L57, L58, L61, L72, L131, L151, L186, L197, L209, L223, L231, L242...

Response: The manuscript has been reviewed again to improve English writing.

-In methodology part, authors should mention the multiple alignment software used in their analysis

Response: Thanks for comment. Relevant software (BioEdit) was added.

-In methodology part, authors should explain why they chose the "Tamura 3-parameter model" in their phylogenetic analysis

Response: Actually, "Tamura 3-parameter model" is this that “Tamura 3-parameter is the analyzing both transitional and transversional rates, G+C content bias, and correcting multiple hits”. This statement has been added to the relevant position in the methodology. 

-Legends of Tables 2, 3 and 4 are misplaced

Response: Modification done!

-A footnote for Table 2 is required to understand the reported data

Response: Footnotes have been added for tables.

-Higher resolution of the figure needs to be provided

Response: Modification done!

Sincerely yours,

Dr. Hamed Mirjalali

Corresponding author

---

## [Decision Letter · Decision Letter 1]

12 Mar 2023

Multigene Typing of Giardia Duodenalis Isolated from Tuberculosis and Non-Tuberculosis Subjects

PONE-D-22-33229R1

Dear Dr. Mirjalali,

We’re pleased to inform you that your manuscript has been judged scientifically suitable for publication and will be formally accepted for publication once it meets all outstanding technical requirements.

Kind regards,

Bibi Razieh Hosseini Farash

Academic Editor

PLOS ONE

Additional Editor Comments (optional):

Reviewers' comments:

Reviewer's Responses to Questions

**Comments to the Author**

1. If the authors have adequately addressed your comments raised in a previous round of review and you feel that this manuscript is now acceptable for publication, you may indicate that here to bypass the “Comments to the Author” section, enter your conflict of interest statement in the “Confidential to Editor” section, and submit your "Accept" recommendation.

Reviewer #1: All comments have been addressed

Reviewer #2: All comments have been addressed

2. Is the manuscript technically sound, and do the data support the conclusions?

Reviewer #1: Yes

Reviewer #2: Yes

3. Has the statistical analysis been performed appropriately and rigorously? 

Reviewer #1: Yes

Reviewer #2: Yes

4. Have the authors made all data underlying the findings in their manuscript fully available?

Reviewer #1: Yes

Reviewer #2: Yes

5. Is the manuscript presented in an intelligible fashion and written in standard English?

Reviewer #1: Yes

Reviewer #2: Yes

6. Review Comments to the Author

Reviewer #1: (No Response)

Reviewer #2: (No Response)

7. PLOS authors have the option to publish the peer review history of their article (what does this mean?). If published, this will include your full peer review and any attached files.

Reviewer #1: No

Reviewer #2: No

---

## [Editor Report · Acceptance letter]

15 Mar 2023

PONE-D-22-33229R1 

Multigene Typing of *Giardia Duodenalis* Isolated from Tuberculosis and Non-Tuberculosis Subjects 

Dear Dr. Mirjalali:

I'm pleased to inform you that your manuscript has been deemed suitable for publication in PLOS ONE. Congratulations! Your manuscript is now with our production department. 

Kind regards, 

on behalf of

Dr. Bibi Razieh Hosseini Farash 

Academic Editor

PLOS ONE